# Molecular Basis of Impaired Decidualization in the Eutopic Endometrium of Endometriosis Patients

**DOI:** 10.3390/cells14050326

**Published:** 2025-02-21

**Authors:** Alejandra Monserrat Retis-Resendiz, Sandra Karen Gómez-Suárez, Elizabeth García-Gómez, Edgar Ricardo Vázquez-Martínez

**Affiliations:** 1Unidad de Investigación en Reproducción Humana, Instituto Nacional de Perinatología (INPer)-Facultad de Química, Universidad Nacional Autónoma de México (UNAM), Mexico City 11000, Mexico; monser9610@gmail.com (A.M.R.-R.); sk-gs@hotmail.com (S.K.G.-S.); 2Secretaría de Ciencia, Humanidades, Tecnologías e Innovación (SECIHTI)-Unidad de Investigación en Reproducción Humana, Instituto Nacional de Perinatología (INPer)-Facultad de Química, Universidad Nacional Autónoma de México (UNAM), Mexico City 11000, Mexico; egarciag1982@gmail.com

**Keywords:** endometriosis, decidualization, eutopic endometrium, endometrial stromal cells, gene expression, signaling pathway, epigenetics

## Abstract

Endometriosis is a chronic gynecological disorder characterized by the presence of endometrial tissue outside the uterine cavity. A common feature of this pathology is the impaired decidualization of endometrial stromal cells, a critical process that prepares the uterus for embryo implantation. This decidualization defect has been mechanistically linked to progesterone resistance in endometriotic lesions. However, the presence and underlying mechanisms of decidualization defects in the eutopic endometrium of women with endometriosis remain controversial. The aim of the present study is to integrate and discuss molecular evidence from both in vivo and in vitro studies examining decidualization alterations in the eutopic endometrium of patients with endometriosis. Multiple studies have demonstrated impaired decidualization in the eutopic endometrium of women with endometriosis. These alterations have been reported on multiple genes, signaling pathways, and epigenetic processes. However, additional functional studies are warranted to elucidate whether these decidualization defects directly contribute to endometriosis-associated infertility. A better understanding of the decidualization process and its dysregulation in endometriosis will not only advance the development of targeted fertility treatments but also facilitate the design of more effective therapeutic strategies for managing this chronic condition.

## 1. Introduction

The endometrium, the innermost layer of the uterus, is essential for reproductive function, particularly during pregnancy [1]. This tissue comprises multiple cell types: luminal and glandular epithelial cells (EECs), stromal cells (ESCs), immune cells such as lymphocytes and macrophages, endothelial cells, smooth muscle cells, and ciliated epithelial cells [2]. In response to cyclic ovarian steroid hormones, the endometrium undergoes dynamic remodeling throughout the menstrual cycle [2,3,4].

The endometrial cycle consists of three distinct phases: proliferative, secretory, and menstrual [5]. During the proliferative phase, the endometrium undergoes estrogen-mediated growth and thickening, accompanied by spiral artery elongation [5,6]. The subsequent secretory phase is characterized by a slight decrease in plasma estradiol (E2) levels and markedly elevated progesterone (P4) levels [5,7,8]. This phase involves enhanced vascular supply, increased mucus secretion, and cessation of proliferation. In addition, the endometrium also undergoes a critical transformation for implantation, a process known as decidualization [7,8]. In non-conception cycles, declining E2 and P4 levels trigger spiral artery constriction, resulting in functional layer ischemia and endometrial shedding during the menstrual phase [5].

Decidualization, critical for pregnancy establishment and maintenance, encompasses functional and morphological modifications that occur within the endometrium to generate the decidua, the specialized tissue supporting pregnancy [7,8]. During this process, ESCs differentiate into specialized secretory cells known as decidual stromal cells (DSCs) in response to post-ovulatory P4 elevation and local cyclic adenosine monophosphate (cAMP) production, involving complex signaling cascades and extensive genetic reprogramming [7,8,9,10].

During decidualization, fibroblast-like ESCs undergo marked morphological changes, becoming enlarged and rounded with prominent nuclei and expanded cytoplasm [8]. Functionally, DSCs acquire a specialized secretory phenotype, producing key implantation factors, including prolactin (PRL) and insulin-like growth factor binding protein-1 (IGFBP1), which serve as decidualization biomarkers [11]. Mounting evidence indicates that cellular or molecular decidualization defects may contribute to pregnancy complications and endometrial disorders, such as endometriosis [12,13,14]. Impaired decidualization has been mainly associated with P4 resistance in endometriotic lesions of endometriosis patients [15,16]. However, decidualization defects in the eutopic endometrium (the normal endometrial tissue located in the proper place in the uterus) of endometriosis patients remain a subject of considerable debate. Evidence suggests that decidualization and implantation-related processes are dysregulated in endometriosis patients [17,18,19]; however, conflicting studies report no significant alterations [17,18,19]. While endometriosis-associated infertility is well documented, the molecular basis of eutopic endometrial dysfunction remains incompletely understood. This review synthesizes current evidence from both in vivo and in vitro studies examining decidualization defects in eutopic endometrium of patients with endometriosis. By integrating recent molecular and cellular findings, we aim to provide comprehensive insights into this underexplored aspect of reproductive biology and identify potential therapeutic targets for improving fertility outcomes in endometriosis patients.

## 2. Symptoms and Pathogenesis of Endometriosis

Endometriosis is an estrogen-dependent inflammatory disorder affecting approximately 10% of reproductive-age women, representing an estimated 176 million women worldwide. This condition is characterized by the presence of endometrial-like tissue outside the uterine cavity (ectopic endometrium), predominantly in reproductive organs, including the ovaries, fallopian tubes, and uterosacral ligaments [20]. Less frequently, lesions may develop in the rectum, bladder, intestines, and vagina [20,21].

Endometriosis manifests with a spectrum of symptoms, including chronic pelvic pain, dyspareunia, and urinary issues; however, symptom severity does not consistently correlate with the extent of lesions [20,21]. This condition is strongly associated with infertility, affecting up to 50% of patients, and notably, endometriosis is implicated in approximately 80% of idiopathic infertility cases [22]. Despite this well-established association between endometriosis and infertility, the underlying biological mechanisms, particularly in mild and moderate disease, remain incompletely understood [22]. In fact, extensive research efforts have yet to fully elucidate the pathogenic mechanisms of endometriosis. Recent reviews describe three main theories regarding its origin: implantation through retrograde menstruation, metaplasia of multipotent mesenchymal stem cells, and lymphatic metastasis of endometrial cells [23].

Similar to the eutopic endometrium, ectopic lesions express receptors for estrogens, P4, and androgens, rendering them responsive to both endogenous and exogenous hormonal stimulation [18,24]. As a result, the ectopic endometrium undergoes cyclic proliferation and breakdown, mirroring the changes observed in the eutopic endometrium; however, its differentiation capacity is reduced [24]. Interestingly, the eutopic endometrium of women with endometriosis exhibits significant alterations in proliferation, adhesion, angiogenesis, and decidualization compared to the endometrium of women without the disease [20,21]. Particularly, impaired decidualization and other endometrial alterations have been associated with P4 resistance and dysregulated P4 receptor (PGR) expression in both eutopic and ectopic endometrial tissues [15,16,25,26,27].

In this review, we clearly differentiate between findings from eutopic and ectopic endometrium in endometriosis. While our primary focus is on eutopic endometrium, we include relevant findings from ectopic lesions to provide a comparative context and highlight the current knowledge gaps regarding eutopic endometrial dysfunction in endometriosis.

## 3. Sex-Steroid Hormonal Signaling, Dysregulation, and Impaired Decidualization in Endometriosis

Since decidualization is a P4-dependent process [8] and the eutopic endometrium of endometriosis patients exhibits P4 resistance [15,28], impaired endometrial function may contribute to the reproductive failure in these patients. This decidualization defect could partially explain the high prevalence of infertility observed in women with endometriosis [20].

The decidualization process is regulated by steroid hormones (SH), primarily E2 and P4, along with other factors that increase intracellular cAMP levels [7,8,9,10]. E2 and P4 exert distinct genomic and non-genomic effects through their respective receptors. Genomic effects are mediated through their respective nuclear steroid receptors (nSR): estrogen receptors (ESR1 and ESR2) [29,30], and PGR [31], which primarily function as transcription factors. Additionally, both hormones can activate membrane receptors, initiating signaling cascades that modulate both nSR-dependent and -independent pathways [31,32]. Particularly, estrogens signal through the membrane G protein-coupled estrogen receptor (GPER, also known as GPR30) [33], while P4 acts via the P4 receptor membrane components (PGRMC) [34] and the class II members of the P4 and adipoQ receptor (PAQR) family [35,36].

PGR has two main isoforms: PGR-A and PGR-B, each regulated by distinct promoter regions and translation start sites [37]. Both isoforms are essential for normal endometrium function, with their expression varying through the menstrual cycle [38]. PGR-A and PGR-B expression is elevated proliferative and early secretory phases but decreases markedly in functional glands during the mid and late secretory phases. In contrast, stromal expression of both isoforms in the functionalis and basalis layers remains constant through the cycle in human endometrial biopsies [38].

Studies using chromatin immunoprecipitation sequencing (ChIP-seq) and RNA sequencing (RNA-seq) in E2- and progestin-treated ESCs have revealed that PGR-B regulates a broader spectrum of endometrial receptivity genes compared to PGR-A, although both isoforms contribute to in vitro decidualization [39].

Initial studies demonstrated that eutopic ESCs isolated from endometriosis patients show impaired in vitro differentiation compared to cells from women without the disease [40]. Additionally, Nisolle and Donnez documented differential expression patterns of ESRs and PGR isoforms between eutopic and ectopic endometrium [40].

Recent evidence demonstrates altered *PGR* expression in the endometrium of endometriosis patients. Attia et al. reported undetectable PGR-B and markedly reduced PGR-A levels in ectopic tissue compared to eutopic endometrium [25]. Igarashi et al. reported a decreased PGR-B: PGR-A ratio in eutopic endometrium from infertile women with endometriosis [41]. Similarly, Shen et al. observed reduced PGR-B protein levels in infertile endometriosis patients compared to infertile controls [42]. Additionally, Wölfler et al. reported decreased *PGR-B* expression during the mid to late secretory phase in the eutopic endometrium of women with endometriosis, suggesting that impaired PGR signaling may contribute to decidualization defects in these patients [43].

*PGR* isoforms expression is regulated by promoter-specific DNA methylation. Rocha et al. demonstrated that while *PGR-A* promoter DNA methylation remains unchanged, the *PGR-B* promoter shows increased DNA methylation levels in the eutopic endometrium from endometriosis infertile patients compared to infertile women without the disease during the secretory phase [44]. This increase in DNA methylation and the decreased *PGR-B* expression reported in previous studies may contribute to impaired endometrial receptivity and function in endometriosis (Figure 1A).

MicroRNAs (miRNAs) are a class of small non-coding regulatory RNAs involved in post-transcriptional gene regulation [45]. miR-194-3p shows elevated expression in the eutopic endometrium from endometriosis patients during the mid-secretory phase. In vitro studies demonstrate that miR-194-3p directly regulates *PGR* expression through interaction with its 3′-untranslated region, with miR-194-3p mimics suppressing and inhibitors enhancing *PGR* levels. Moreover, miR-194-3p overexpression impairs ESC decidualization [46], suggesting that this miRNA contributes to P4 resistance and decidualization defects in the eutopic endometrium of women with endometriosis.

Vázquez-Martínez et al. reported reduced *PAQR7* and *PAQR5* in both eutopic and ectopic endometrium from endometriosis patients compared to control endometrium. In contrast, *PAQR8* and *PAQR6* expression was decreased exclusively in the eutopic endometrium. However, reduced PAQR7 and PAQR8 protein levels were only observed in the ectopic endometrium [36]. This downregulation of PAQRs in endometriosis may contribute to the P4 resistance observed in these patients (Figure 1B). Further studies are needed to determine whether PAQR expression is dysregulated in the eutopic endometrium of infertile endometriosis patients.

On the other hand, endometriosis is an estrogen-dependent disorder, with E2 being essential for ectopic tissue [20]. In endometriosis women, E2 reaches both eutopic and ectopic endometrium through circulation but is predominantly synthetized locally due to elevated expression of key steroidogenic enzymes: aromatase (CYP19A1, the rate-limiting enzyme in estrogen biosynthesis) and steroidogenic acute regulatory protein (StAR, mediates cholesterol transport to the inner mitochondrial membrane). This local steroidogenic capacity contrasts with normal endometrium, which shows almost undetectable levels of these biosynthetic enzymes [47,48]. In vitro studies demonstrated that 8-Br-cAMP treatment of ESCs from the eutopic endometrium of endometriosis patients increases CYP19A1 expression while decreasing HSD3B1 (which encodes the pregnenolone to P4 converting enzyme). Additionally, 8-Br-cAMP upregulated HSD17B1 (encoding the estrone to E2 converting enzyme) in endometriotic ESCs [49]. These alterations in steroidogenic enzyme expression promote an estrogen-dominant environment within the endometrium, potentially contributing to impaired decidualization in endometriosis.

Local E2 accumulation contributes to lesion development and progression through the activation of ESRs [50]. In healthy endometrium, ESR1 expression shows cyclic variation, with higher levels during the proliferative phase compared to the secretory phase (in both functional and basalis stroma), while ESR2 expression remains relatively constant throughout the menstrual cycle [38]. This pattern, combined with significantly higher ESR1 expression compared to ESR2, suggests a predominant role for ESR1 in mediating endometrial estrogen responses [51].

*ESR1* and *ESR2* expression is upregulated in the eutopic and ectopic endometrium from endometriosis patients compared to control endometrium [52], with ectopic lesions showing higher expression of both genes compared to eutopic tissue [51]. Dysregulation of the ESR2/ESR1 ratio in ectopic endometrial stromal cells has been implicated in endometriosis pathogenesis and disease severity [53] (Figure 1C). Further investigation of the ESR2/ESR2 ratio in eutopic endometrium from endometriosis patients may reveal its potential role in disease pathogenesis and progression.

An enhanced understanding of endometriosis-associated molecular mechanisms provides crucial insights for developing hormone signaling-targeted therapies. Specifically, elucidating progesterone resistance mechanisms will facilitate the development of selective progesterone receptor modulators that account for PGR isoform-specific alterations and epigenetic regulation of receptor expression. Novel therapeutic strategies targeting membrane progesterone receptors could include PAQR family-specific modulators and pathway-selective interventions. Additionally, interventions aimed at normalizing estrogen signaling through ESR2/ESR1 ratio modulation, regulation of local estrogen production, and integration with progesterone signaling pathways represent promising therapeutic approaches. These mechanistic insights will enable the development of more targeted and effective treatments for endometriosis-associated infertility.

## 4. Dysregulation of *IGFBP1* and *PRL* Expression During Decidualization in Endometriosis

Multiple molecular and genetic studies have identified aberrant regulation of decidualization pathways in endometriosis. In this section, we focus on *IGFBP1* and *PRL* expression dysregulation during the decidualization of ESCs from patients with endometriosis (Table 1).

Minici et al. demonstrated reduced decidualization capacity in eutopic ESCs from endometriosis patients, evidenced by significantly reduced IGFBP1 and PRL secretion compared to control ESCs following E2 and 6α-methyl-17α-hydroxyprogesterone acetate (MPA) treatment for 7 and 13 days [17]. Although extended treatment (16 days) resulted in increased PRL secretion compared to controls, suggesting potential recovery of decidualization capacity in vitro [17], these findings must be interpreted cautiously given the absence of in vivo regulatory factors. Notably, treatment of control ESCs with peritoneal fluid from endometriosis patients (EPF), characterized by elevated cytokines, growth factors, and adhesion molecules, reproduced the decidualization defects [17]. Interestingly, when control ESCs were treated with EPF, they showed similar defects in decidualization markers. This suggests that both intrinsic molecular alterations and the inflammatory microenvironment contribute to impaired endometrial function in endometriosis.

Klemmt et al. further demonstrated reduced decidualization capacity in both ectopic and eutopic ESCs from endometriosis patients, with decreased PRL and IGFBP1 secretion following 8-Bromo-cAMP (8-Br-cAMP) treatment [24]. Similarly, Barragán et al. reported undetectable IGFBP1 secretion in endometrial mesenchymal stem cells (eMSCs) and endometrial stromal fibroblasts progenitors (eSFs) from endometriosis patients following a 14-day E2 and P4 treatment, contrasting with significant IGFBP1 secretion in cells derived from women without endometriosis [19]. Studies using menstrual effluent-derived stromal fibroblast cells (dME-SFCs) derived from endometriosis patients and controls have demonstrated reduced IGFBP1 secretion in endometriosis group at 6, 24, and 48 h post-8-Br-cAMP stimulation [54], with similar findings reported by Nayyar et al. at 24 h [55]. Single-cell RNA sequencing analysis by Shih et al. revealed significant *IGFBP1* expression in unstimulated control dME-SFCs compared to endometriotic cells [56].

In ESCs isolated from both endometriosis patients and controls, P4 treatment (14 days) upregulates *IGFBP1* and *PRL* expression. However, when treated with 8-Br-cAMP (96 h), control ESCs show significantly higher expression of both markers compared to endometriotic ESCs [49], suggesting PKA pathway involvement in this differential response. Similarly, Su et al. demonstrated that combined E2, P4, and dibutryl-cAMP (db-cAMP) treatment for 8 days resulted in significantly higher *IGFBP1* and *PRL* expression in control compared to eutopic (endometriosis) ESCs [57]. Tsuno et al. reported that while *PRL* and *IGFBP1* expression remained unchanged in eutopic ESCs from women with endometriosis treated with db-cAMP and medroxyprogesterone (MPA) or dienogest (12 days), ectopic ESCs showed significantly lower expression compared to both eutopic (endometriosis) and control cells [58]. These variable results highlight the need for standardized in vitro decidualization protocols that better reflect physiological conditions.

Collectively, these studies demonstrate impaired decidualization capacity in eutopic ESCs from endometriosis patients, characterized by reduced expression and secretion of the decidualization markers PRL and IGFBP1. While more studies on eutopic (endometriosis) ESCs and standardized protocols are needed for more reproducible results, current evidence indicates that both intrinsic molecular alterations and environmental factors contribute to defective decidualization in the endometrium of women with endometriosis.

**Table 1 cells-14-00326-t001:** Effects of different in vitro decidualization treatments on IGFBP1 and PRL secretion/expression in eutopic and control * endometrial stromal cells.

Cells Type	Treatment	Days of Treatment	Results	References
ESCs	1 × 10^−8^ M E2 2 × 10^−7^ M MPA	6 and 13 days	Decreased secretion of IGFBP1 and PRL in eutopic endometrium compared to control.	Minici et al. [17]
		16 days	Increased secretion of PRL in eutopic endometrium compared to control.	
ESCs	5 × 10^−4^ M 8-Br-cAMP	3 to 20 days	Decreased secretion of IGFBP1 and PRL in eutopic endometrium compared to control.	Klemmt et al. [24]
eMSC and eSF	1 × 10^−8^ M E2 1 × 10^−6^ M P4	14 days	Undetectable expression of IGFBP1 in eutopic endometrium.	Barragan et al. [19]
ESCs	1 × 10^−6^ M P4	14 days	Increased expression of *IGFBP1* and *PRL* in both eutopic and control endometrium.	Aghajanova et al. [49]
	5 × 10^−4^ M 8-Br-cAMP	14 days	Decreased expression of *IGFBP1* and *PRL* in eutopic endometrium compared to control.	
ESCs	3.6 × 10^−8^ M E2 1 × 10^−6^ M P4 1 × 10^−4^ M db-cAMP	6 and 8 days	Decreased expression of *IGFBP1* and *PRL* in eutopic endometrium compared to control.	Su et al. [57]
ESCs	5 × 10^−4^ M 8-Br-cAMP 1 × 10^−6^ M MPA/dienogest	12 days	No significant changes in *IGFBP1* and *PRL* expression between eutopic and control endometrium.	Tsuno et al. [58]
dME-SFCs	5 × 10^−4^ M 8-Br-cAMP	6, 24 and 48 h	Decreased secretion of IGFBP1 in eutopic compared to control.	Warren et al. [54]
dME-SFCs	5 × 10^−4^ M 8-Br-cAMP	24 h	Decreased secretion of IGFBP1 in eutopic compared to control.	Nayyar et al. [55]

* Eutopic endometrium refers to the endometrium properly located in the uterus in patients with endometriosis, and the endometrium from women without the disease is referred to control endometrium.

## 5. Molecular and Cellular Disruptions of Decidualization in Endometriosis: Altered Gene Expression, Pathway Dysregulation, and Inflammatory Influences

Klemmt et al. demonstrated delayed morphological transition during in vitro decidualization in eutopic ESCs from endometriosis patients, requiring 6 days of 8-Br-cAMP treatment compared to 3 days in ESCs from women without the disease [24,59], supporting the presence of molecular defects affecting decidualization capacity. This altered temporal response may have significant implications, given that successful implantation requires precise temporal regulation of endometrial receptivity.

Recent molecular analyses revealed that E2, P4, and cAMP-stimulated eutopic ESCs from endometriosis patients exhibit distinct transcriptional signatures compared to control cells [59]. Major dysregulation was observed in the TGFβ signaling pathway and its key regulators, including SMAD4 and bone morphogenetic proteins (BMPs) [59]. Typically, BMPs signal through SMAD1/5/4 and TGFβ through SMAD2/3/4 [60]. In this context, eutopic ESCs show altered SMAD4 and H3K27ac binding patterns during decidualization (Figure 2). These findings suggest that impaired TGFβ-mediated transcriptional responses are SMAD4-dependent. Notably, exogenous BMP2 treatment restored the expression of the downregulated decidualization markers IGFBP1 and FOXO1 in both eutopic (endometriosis) ESCs and epithelial assembloids [59].

FOXO1, a transcription factor that regulates cellular metabolism, cell-cycle progression, oxidative stress response, and apoptosis [61], plays a crucial role in decidualization, such as the regulation of *IGFBP1* and *PRL* expression [62]. The eutopic endometrium of endometriosis patients exhibit reduced *FOXO1* expression compared to controls, suggesting that its dysregulation contributes to impaired decidualization [57]

In endometriosis, multiple pathways converge to disrupt FOXO1 function. Su et al. demonstrated downregulation of Notch signaling pathway genes (*NOTCH1*, *NOTCH4*, *JAGGED2*, and *DLL4*) in both the eutopic endometrium and in vitro decidualized eutopic ESCs of endometriosis patients. Moreover, *NOTCH1* silencing reduced *FOXO1* expression more markedly in ESCs from endometriosis patients compared to controls [57], indicating that impaired Notch signaling contributes to decidualization defects through FOXO1 downregulation (Figure 2). Kang et al. revealed elevated levels of phosphorylated AKT1 (pAKT1), Calpain-7 (CAPN7), and phosphorylated FOXO1 (Ser319) in the endometrium of endometriosis patients compared to controls. Their findings suggest that CAPN7 promotes AKT1 phosphorylation, leading to FOXO1 phosphorylation and nuclear exclusion, thereby inhibiting decidualization (Figure 2B) [63]. Additionally, NEK2 is upregulated in both eutopic and ectopic endometrium from endometriosis patients compared to controls, correlating with decreased FOXO1 levels [64]. NEK2 phosphorylates FOXO1 at Ser184, reducing protein stability and promoting cell proliferation while impairing decidualization (Figure 2B) [64].

Collectively, these studies demonstrate that reduced FOXO1 expression and the associated defective decidualization in eutopic endometrium of endometriosis patients results from multiple dysregulated pathways, including impaired Notch signaling and enhanced CAPN7 and NEK activity [64].

Connexin 43 (Cx43), a gap junction protein essential for cellular communication, plays a critical role in endometrial receptivity and decidualization through the regulation of angiogenesis [65,66]. Yu et al. demonstrated reduced Cx43 expression in the eutopic endometrium in endometriosis patients. Furthermore, E2, P4, and cAMP-induced decidualization of eutopic ESCs from endometriosis patients for 7 days showed impaired epithelioid transformation and significantly reduced Cx43 protein upregulation compared to controls [67]. This gap junction dysfunction may disrupt intercellular transport of essential small metabolites (<1 kDa), potentially affecting DSC morphological transformation and endometrial receptivity through altered angiogenic processes [65,66]. Interestingly, under the previously mentioned decidualization protocol, IL-1β suppresses Cx43, PRL, and VEGF expression through ERK1/2 and p38 MAP kinase pathways [68]. Notably, the eutopic endometrium of women with endometriosis exhibits elevated levels of inflammatory cytokines, including IL-1β, TNF-α, and IL-8 compared to the control [69], suggesting that this pro-inflammatory environment contributes to impaired decidualization in endometriosis.

IL-11 expression is reduced in the eutopic endometrium of endometriosis patients compared to controls [70]. It has been demonstrated that IL-11 plays essential roles in decidualization in both mouse and human systems, although its regulation differs between species [71,72]. In mice, IL-11 expression increases post-implantation, while in humans, P4 drives its upregulation during the secretory phase of the menstrual cycle [73,74]. Despite these regulatory differences, IL-11 dysregulation is associated with impaired decidualization in both species [74]. Karpovich et al. demonstrated reduced IL-11 levels in eutopic ESCs during in vitro decidualization with 8-Br-cAMP compared to controls, correlating with decreased PRL and IGFBP1 levels [75]. This reduction in IL-11 expression in the eutopic endometrium, likely resulting from progesterone resistance and altered PKA signaling, may contribute to endometriosis-associated infertility through multiple mechanisms: impaired decidualization, defective implantation, compromised placentation, altered Natural Killer cell differentiation, and decreased decidual cell survival [75,76,77,78,79,80,81]. However, the precise mechanisms linking IL-11 signaling to decidualization in the eutopic endometrium of endometriosis patients require further investigation [17].

As previously established, environmental factors may also contribute to impaired decidualization in endometriosis. Minici et al. demonstrated that EPF suppresses PRL secretion in decidualizing ESCs from controls treated with E2+MPA for 16 days [17]. Given the elevated TNF-α levels in EPF [81], the addition of soluble TNF-α receptor 1 (sTNFR-1) partially rescued PRL secretion in eutopic (endometriosis) ESCs, suggesting that TNF-α mediates the inhibitory effects of EPF [17].

Barragan et al. also reported that fibroblasts of ESCs from the eutopic endometrium of endometriosis patients exhibit a distinct pro-inflammatory transcriptional profile compared to controls [19]. Collectively, these studies demonstrate that both impaired Cx43 expression and elevated pro-inflammatory mediators contribute to decidualization defects in endometriosis. These molecular alterations represent potential therapeutic targets, though additional research is needed to fully characterize the pathways involved and develop targeted interventions.

Tiberi et al. demonstrated reduced expression of prokineticin (PROK1), a critical mediator of implantation and decidualization, in the endometrium of women with endometriosis compared to controls [82]. They also reported that during in vitro decidualization with E2+MPA, gene expression of both *PROK1* and *PGR* was undetectable in eutopic (endometriosis) ESCs at 72 and 96 h, while control ESCs showed significant upregulation. However, after extended treatment (16 days), gene expression patterns were similar in both groups [82]. These findings support the idea that eutopic ESCs from endometriosis patients display an initial resistance to decidualization signals but can eventually respond to prolonged hormonal stimulation.

## 6. Epigenetic Mechanisms and Their Role in Endometrial Dysfunction in Endometriosis

Epigenetics encompasses heritable modifications in gene expression mediated through alterations in chromatin structure without changes in DNA sequence [83]. These regulatory mechanisms operate through DNA base modifications, chromosomal remodeling, and modifications at multiple molecular levels, influenced by both genetic and environmental factors. These modifications can be transmitted across generations and are implicated in various pathological conditions [83,84].

Recent studies have focused on characterizing epigenetic dysregulation in endometriosis. Here, we examine its contribution to impaired decidualization (Figure 3 and Table 2). The participation of DNA methylation in the regulation of *PGR* isoforms expression was described in the “Hormonal Signaling, Dysregulation and Impaired Decidualization in Endometriosis” section [39,41,43,44,46,85,86,87].

Histone modifications regulate gene expression through alterations in chromatin structure and nucleosome dynamics [88]. Eutopic and ectopic endometrium exhibit distinct histone methylation patterns at H3K4, H3K9, and H3K27 compared to control endometrium [89,90,91]. These modifications show menstrual cycle-dependent changes that may contribute to impaired decidualization [92] and endometriosis-associated infertility.

Zang et al. demonstrated elevated H3K27me3 and H3K9me3 levels in both ectopic and eutopic endometrium of endometriosis patients compared to control endometrium [89]. These epigenetic alterations show lesion-specific patterns, with deep infiltrating endometriosis (DIE) exhibiting higher levels of EZH2, H3K27me3, and H3K9me3 compared to ovarian endometriomas (OMA), suggesting progressive epigenetic reprogramming [93]. However, the role of these histone modifications in the decidualization of eutopic ESCs of women with endometriosis remains to be elucidated.

In a healthy endometrium, EZH2 levels progressively decrease throughout the menstrual cycle. This reduction is paralleled in ESCs during in vitro decidualization, accompanied by decreased H3K27me3 enrichment at *PRL* and *IGFBP1* transcription start sites in response to 8-Br-cAMP and MPA [94]. Simultaneously, H3K27ac enrichment increases at a distal enhancer region upstream of *IGFBP1*, modulating its expression [95]. *IGFBP1* upregulation involves cAMP-mediated recruitment of C/EBPβ, FOXO1, and p300 to its enhancer region. Notably, C/EBPβ inhibition disrupts multiple regulatory mechanisms, including H3K27ac modification, chromatin remodeling, and p300 recruitment at the *IGFBP1* enhancer [95,96].

Endometriotic lesions exhibit elevated EZH2 and H3K27me3 levels [93], a characteristic influenced by the hypoxic microenvironment. Hypoxia induces *EZH2* expression in eutopic (endometriosis) ESCs, contributing to impaired decidualization [97]. Lin et al. demonstrated that in normal ESCs, the N6-Methyladenosine (m^6^A) reader YTHDF2 targets *EZH2* mRNA for degradation, reducing H3K27me3 levels at the *IGFBP1* promoter and enabling *IGFBP1* expression during decidualization. However, hypoxia upregulates the m^6^A demethylase ALKBH5, which stabilizes *EZH2* mRNA by removing m^6^A modifications, consequently suppressing *IGFBP1* expression. Supporting these findings, mouse studies demonstrate that *Ezh2* depletion enhances endometrial receptivity and fertility through *IGFBP1* upregulation [98].

Prostaglandin E2 (PGE2) signaling through EP2 and EP4 receptors cooperates with steroid hormones via PKA signaling to induce decidualization in both human ESCs and rodent models [10,99,100,101,102,103,104]. In endometriosis patients, PGE2 levels are elevated in peritoneal fluid [105], with increased PGE2 production and cyclooxygenase-2 (COX-2) expression in both ectopic and eutopic endometrium compared to healthy endometrium [106]. Remarkably, selective COX-2 inhibitors reduce endometriosis-associated pain [107,108,109]. EP2 and EP4 receptors are expressed in various endometriosis lesions (ovarian, adenomyotic, and peritoneal), and their specific antagonists also alleviate endometriosis-associated pain [110,111]. The eutopic endometrium of endometriosis patients exhibits increased expression of COX-2, EP2, and EP4 compared to controls [112]. Notably, selective EP2/EP4 antagonists suppress cAMP signaling, proliferation, proinflammatory cytokine production, and aromatase expression in endometriotic ESCs [110].

Arosh et al. demonstrated that pharmacological inhibition of EP2/EP4 in endometriotic cells decreases H3K9 and H3K27 methylation while increasing their acetylation [113]. Additionally, EP2/EP4 inhibition selectively increases H3K4 methylation in stromal but not epithelial endometriotic cells [113]. These findings suggest that PGE2 signaling modulates chromatin state in endometriosis, leading to the establishment of aberrant euchromatin states. The complex interaction between PGE2, proinflammatory cytokines, estrogen synthesis, and lesion innervation identify EP receptors as promising therapeutic targets. While these pathways likely impact fertility in endometriosis patients, their precise role in eutopic endometrial decidualization requires further investigation.

Key regulatory genes (*MLL1*, *COX4/2*, *HOXA10*, *PRMT5*) show reduced expression compared to the normal endometrium. *HOXA10* regulation is particularly altered during the secretory phase, characterized by increased MeCP2 occupancy and a transcriptional repressive chromatin state at its promoter, resulting in transcriptional repression.

Epigenetic regulation requires precise coordination of histone-modifying enzymes, including histone lysine methyltransferases (KMTs) [114], histone acetyltransferases (HATs) [115], or histone deacetylases (HDACs). The dysregulation of these enzymes has been implicated in various pathologies, including endometrial cancer and endometriosis.

Mixed lineage leukemia 1 (MLL1), which catalyzes H3K4 trimethylation at gene promoters [116], shows endometrial cycle-dependent expression in the normal endometrium but is downregulated in the eutopic endometrium of endometriosis patients, correlating with reduced H3K4me3 levels [90]. During in vitro decidualization, *MLL1* expression and H3K4me3 levels increase in control ESCs, an effect blocked by the P4 antagonist mifepristone (PGR antagonist). *MLL1* silencing prevents *IGFBP1* and *PRL* induction and inhibits characteristic decidual morphological changes, demonstrating its essential role in decidualization [117]. Additionally, PGR-P4 complexes bind directly to the *MLL1* promoter, demonstrating transcriptional regulation of *MLL1* by P4 signaling. In the eutopic endometrium of endometriosis patients, decreased *PGR* expression correlates with reduced MLL1 levels, resulting in diminished H3K4me3 enrichment at promoters of decidualization-related genes [117].

Cytochrome c oxidase COX4/2, normally upregulated during the secretory phase, remains suppressed throughout the menstrual cycle in the eutopic endometrium of patients with endometriosis [117]. In vitro decidualization induces MLL1 recruitment to the *COX4/2* promoter, increasing H3K4me3 enrichment in immortalized ESCs. *MLL1* silencing prevents both MLL1 recruitment and H3K4me3 enrichment, correlating with COX4/2 downregulation in endometriosis [117]. This dysregulation may promote a hypoxic endometrial microenvironment through reduced COX4/2-mediated oxygen affinity.

Cai et al. demonstrated reduced expression of protein arginine methyltransferase 5 (PRMT5) in the eutopic endometrium (only in the stroma compartment) of endometriosis patients compared to fertile controls during the mid-secretory phase of the menstrual cycle. While *PRMT5* expression increases during normal in vitro decidualization, pharmacological inhibition with GSK591 prevents *IGFBP1* induction and PRL secretion, establishing PRMT5’s essential role in ESC decidualization. Transcriptomic analysis revealed that decreased PRMT5 activity enhances NF-κB signaling through nuclear p65 translocation, a feature observed in endometriotic tissue. Notably, *PRMT5* overexpression restores *IGFBP1* and *PRL* expression in eutopic ESCs of endometriosis patients [118], suggesting that PRMT5 dysregulation contributes to impaired decidualization in endometriosis. These findings identify histone-modifying enzymes as potential therapeutic targets for improving decidualization in endometriosis-associated infertility.

**Table 2 cells-14-00326-t002:** Epigenetic alterations in the eutopic endometrial cells of endometriosis patients.

Epigenetic Mechanism	Alteration in Endometriosis	Effect on Gene Expression and Decidualization	Reference
DNA methylation of *PGR-B* Promoter	Increased methylation in eutopic endometrium during the secretory phase in endometriosis	Associated with decreased *PGR-B* expression, potentially affecting decidualization and P4 responsiveness	Rocha-Junior et al. [44]
H3K27me3 and H3K9me3 levels	Significantly higher levels in eutopic endometrium from endometriosis patients	Not reported	Zhang et al. [89]
EZH2, H3K27me3, and m^6^A levels	Hypoxia induces ALKBH5 that promotes *EZH2* mRNA stability by removing m^6^A and increases H3K27me3 levels.	Increased EZH2 and H3K27me3 levels that downregulate *IGFBP1* expression, impairing decidualization and contributing to eutopic endometrial dysfunction in endometriosis	Lin et al. [98]
MLL1 and H3K4me3 levels	Decreased *MLL1* and H3K4me3 levels in the eutopic endometrium of endometriosis patients	Possibly associated with lower enrichment of H3K4me3 in promoter regions of decidualization-associated genes such as *COX4/2*	Wen et al. [117]
HDAC3 levels	Lower HDAC3 levels in the endometrium of infertile women with endometriosis	Reduced HDAC3 levels may contribute to aberrant gene expression of decidualization-associated genes such as *IGFBP1* and *PRL*	Kim et al. [119]
SIRT1 levels	Significantly higher levels of SIRT1 in both eutopic ESCs and EECs of endometriosis patients.	Overexpression of SIRT1 may lead to suppressed *IGFBP1* and *PRL* expression during decidualization	Kim et al. [120] Yu et al. [121]
H3K9me3 and H3K9ac levels on *HOXA10* promoter	Decreased expression of *HOXA10* is associated with increased H3K9me3 and reduced H3K9ac levels in eutopic endometrium from endometriosis patients	Reduced HOXA10 expression may impair endometrial receptivity and decidualization, contributing to infertility in endometriosis patients	Samadieh et al. [122] Gui et al. [123]
PCAF levels	Abnormal expression of *PCAF* in eutopic endometrium of endometriosis patients	Reduced PCAF activity leads to decreased acetylation of HOXA10-regulated genes, impairing gene expression needed for endometrial receptivity and function.	Zhu et al. [124]

HDAC3 shows reduced protein levels in the eutopic endometrium of infertile women with endometriosis compared to controls. *HDAC3* silencing in ESCs from women without endometriosis impairs in vitro decidualization, evidenced by decreased *IGFBP1* and *PRL* expression. Additionally, while *COL1A1* and *COL1A2* are normally downregulated during decidualization, HDAC3 silencing induces their expression [119], suggesting that HDAC3 deficiency contributes to decidualization defects in endometriosis.

SIRT1, another histone modifier, is significantly upregulated in both the eutopic ESCs and EECs of endometriosis patients compared to controls [120,121]. While normal decidualization reduces SIRT1 expression, its overexpression in T-hESC suppresses *PRL* and *IGFBP1* expression [121]. These findings indicate that elevated SIRT1 levels in the eutopic endometrium of endometriosis patients may contribute to impaired decidualization and disease pathogenesis.

HOXA10, a key regulator of uterine development and endometrial differentiation, shows altered expression in endometriosis [125,126,127,128,129]. In healthy endometrium, *HOXA10* expression is regulated by sexual steroid hormones [130] and exhibits cycle-dependent patterns [131], with significant upregulation during the secretory phase correlating with elevated E2 and P4 levels. However, this physiological increase is absent in endometriosis [123,129]. This reduced *HOXA10* expression in the eutopic endometrium during the secretory phase correlates with specific epigenetic alterations: decreased H3K9 acetylation, increased H3K9 methylation, and enhanced MeCP2 enrichment at the *HOXA10* promoter [122]. Furthermore, dysregulation of p300/CREB-binding protein-associated factor (PCAF) impairs HOXA10-mediated gene regulation through altered acetylation patterns in the eutopic endometrium of women with endometriosis [124].

Together, these studies suggest that several epigenetic modifications contribute to impaired endometrial receptivity and decidualization in endometriosis. Further functional studies are needed to validate these findings and investigate additional histone modifications and epigenetic mechanisms in eutopic endometrium, as current research has primarily focused on ectopic lesions. Understanding these molecular mechanisms will suggest several promising therapeutic strategies, including targeted epigenetic interventions to modulate DNA methylation, the regulation of histone-modifying enzymes, and the development of selective EP2/EP4 receptor antagonists to normalize histone modifications. Future therapeutic approaches could incorporate tissue-specific delivery systems to maximize efficacy while minimizing off-target effects.

## 7. Therapeutic Approaches Targeting Dysregulated Pathways in the Eutopic Endometrium of Endometriosis Patients

As epigenetic changes are reversible, developing therapies targeting enzymes or molecules involved in altering methylome and histone patterns seems a viable alternative for treating diseases such as endometriosis [132], particularly as our understanding of epigenetic regulation in this disease continues to expand.

While clinical trials targeting the molecular pathways described in this review are currently lacking, some drugs have potential as epigenetic therapeutics for endometriosis-associated infertility.

Nucleoside and non-nucleoside DNMT inhibitors, which provide sustained epigenetic modulation, represent promising therapeutic candidates [133]. Specifically, selective DNMT3B inhibitors have been proposed as potential therapeutic targets for endometriosis treatment [133]. While further investigation in the context of endometriosis is needed, structure-based drug design approaches targeting DNMT3B have already identified promising compounds for cancer therapy, suggesting potential applications for other epigenetic disorders [134]. However, potential adverse effects must be considered, as demethylating agents such as 5-aza-20-deoxycytidine (5-Aza-dC) increase aromatase expression in patient-derived endometrial stromal cells, potentially promoting endometriotic-like characteristics [135].

HDAC inhibitors represent another promising therapeutic approach for endometriosis. These compounds modulate gene expression in endometriotic cyst stromal cells, inhibiting cell proliferation and inducing cell cycle arrest and apoptosis [136]. Specific HDAC1 inhibitors, particularly benzamides like Chinamid, warrant further investigation as potential therapeutic agents [137].

While various compounds have been evaluated in preclinical and clinical studies, most current therapeutic approaches for endometriosis focus on suppressing ectopic lesions or managing pain symptoms [138]. However, GnRH antagonists (Elagolix) and aromatase inhibitors (Letrozole) have emerged as promising treatments specifically addressing endometriosis-associated infertility [139]. Nevertheless, further research is required to develop targeted therapies that address the molecular mechanisms underlying decidualization defects in the eutopic endometrium.

## 8. Conclusions and Perspectives

Current evidence demonstrates that endometriosis is characterized by profound alterations in the decidualization of eutopic endometrium, involving dysregulation of steroid hormone signaling, transcriptional networks, and epigenetic mechanisms. The eutopic endometrium of endometriosis patients exhibits reduced *PGR* expression, particularly *PGR-B*. This progesterone resistance is further compounded by alterations in membrane progesterone receptors and disrupted estrogen signaling.

Decidualization defects are evidenced by impaired expression of key biomarkers such as IGFBP1 and PRL, altered cellular morphology, and dysregulated inflammatory responses. These abnormalities involve multiple pathways, including disrupted TGFβ signaling, altered gap junction communication through Cx43, and compromised epigenetic regulation through histone-modifying enzymes such as MLL1, PRMT5, and HDAC3.

While significant progress has been made in understanding decidualization defects in endometriosis, several key areas require further investigation, such as the comprehensive molecular characterization using advanced omics approaches, functional studies linking decidualization impairment to infertility, standardization of in vitro decidualization protocols, and investigation of potential therapeutic targets.

An enhanced understanding of these mechanisms will be crucial for developing targeted therapies to improve both fertility outcomes and disease management in endometriosis patients.

## Figures and Tables

**Figure 1 cells-14-00326-f001:**
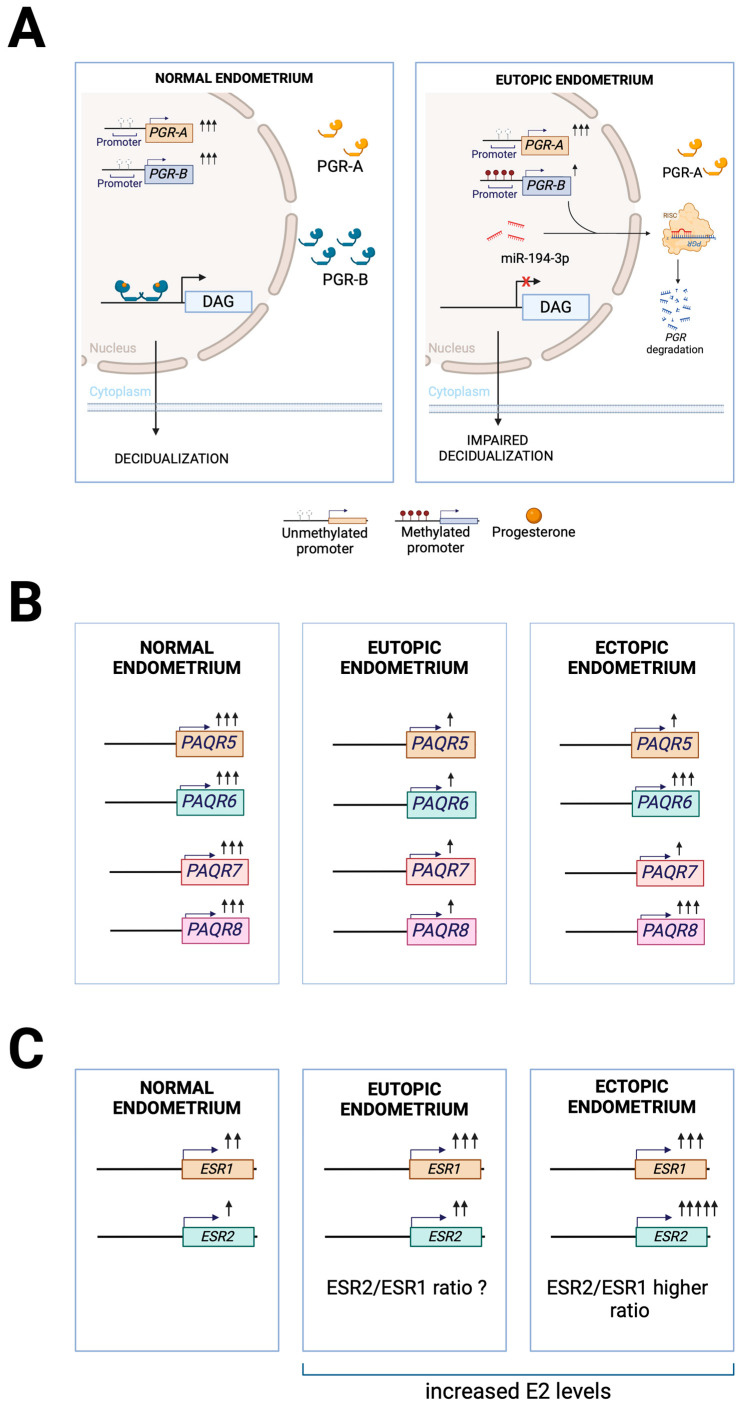
Molecular Alterations in Sex-Steroid Hormone Receptors and Signaling in Endometriosis. (**A**) Progesterone receptor (PGR) signaling in the normal and eutopic endometrium of endometriosis patients. In a healthy endometrium, PGR-B and PGR-A isoforms are expressed, and ligand-bound PGR-B regulates the expression of decidualization-associated genes (DAGs). In endometriosis, *PGR-B* promoter hypermethylation and miR-194-3 mediated degradation reduce its protein levels, impairing decidualization. (**B**) Membrane progesterone receptor (PAQR) expression patterns across endometrial tissues. Differential expression of membrane progesterone receptors in normal endometrium compared to eutopic and ectopic endometriotic tissues. Expression levels are represented by the number of arrows. (**C**) Estrogen receptor dysregulation in endometriosis. *ESR1* and *ESR2* show increased expression in both eutopic and ectopic endometrial tissues in endometriosis patients compared to control endometrium, with ectopic lesions exhibiting the highest ESR2/ESR1 ratio. However, whether the ESR2/ESR1 ratio is altered in eutopic endometrium remains to be determined.

**Figure 2 cells-14-00326-f002:**
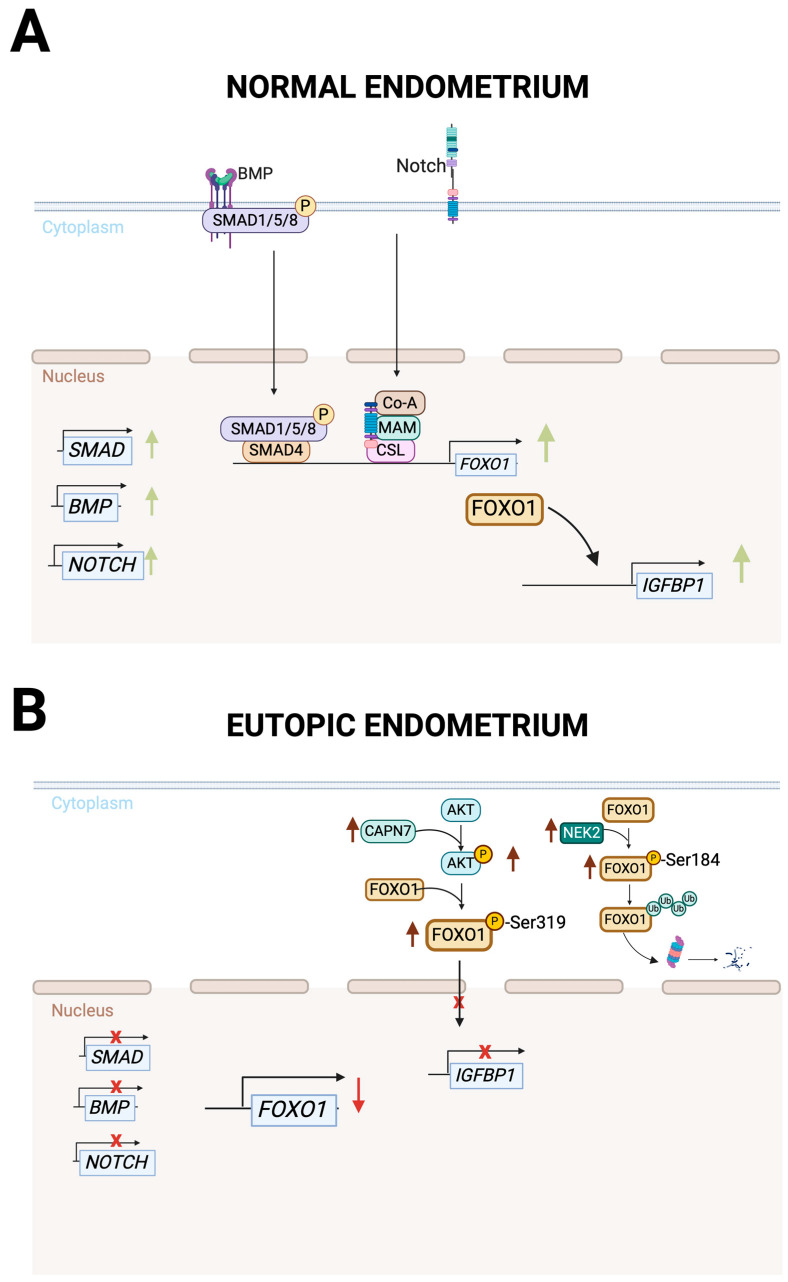
FOXO1 Regulation in the Normal and Eutopic Endometrium of Endometriosis Patients. Signaling pathways involved in the regulation of FOXO1 during in vitro decidualization in ESCs from control endometrium (**A**) and from eutopic endometrium of endometriosis patients (**B**). (**A**) During in vitro decidualization, *BMP*, *SMAD*, and *NOTCH* genes are upregulated. BMP-SMAD and Notch signaling independently promote *FOXO1* transcription through SMAD and NICD (Notch Intracellular Domain) activation, respectively. FOXO1 then regulates decidualization markers, including *IGFBP1*, enabling normal endometrial function. (**B**) Multiple pathway dysregulation reduces FOXO1 levels during decidualization in the eutopic endometrium of endometriosis patients. Downregulation of SMAD-BMP and Notch signaling decreases *FOXO1* expression. Enhanced CAPN7 and pAKT1 activity promotes FOXO1 phosphorylation (Ser319), triggering nuclear exclusion. Overexpression of NEK2 induces FOXO1 phosphorylation (Ser184) that, in turn, promotes ubiquitination and proteasomal degradation, ultimately impairing decidualization.

**Figure 3 cells-14-00326-f003:**
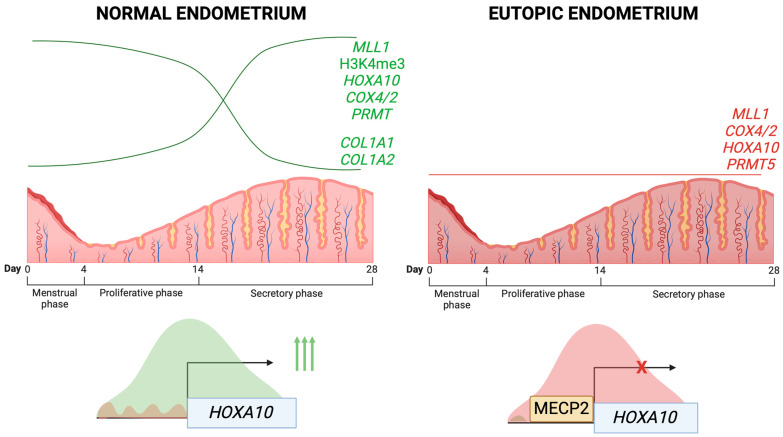
Epigenetic regulation of gene expression in normal and eutopic endometrium tissues of endometriosis patients. The menstrual cycle is divided into three phases: menstrual, proliferative, and secretory. Normal endometrium (**left panel**): *MLL1*, *HOXA10*, *COX4/2*, *PRMT5*, and H3K4me3 levels show progressive upregulation during the menstrual cycle, while *COL1A1* and *COL1A2* maintain low expression levels. During the secretory phase, *HOXA10* expression is enhanced by a transcriptional active chromatin state at its promoter. Eutopic endometrium from endometriosis patients (**right panel**).

## Data Availability

No new data were created or analyzed in this study. Data sharing is not applicable to this article.

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
