# Peer review of "Molecular Basis of Impaired Decidualization in the Eutopic Endometrium of Endometriosis Patients"

_cells, 2025, doi:10.3390/cells14050326_

Round 1

Reviewer 1 Report

Comments and Suggestions for Authors

The manuscript titled "Molecular Basis of Impaired Decidualization in Eutopic Endometrium of Endometriosis Patients" is a well-structured and detailed review of an important and underexplored topic in reproductive biology. The authors have done an excellent job of integrating a broad range of current literature, presenting complex mechanisms in an accessible manner, and offering clinical and therapeutic perspectives. However, while the manuscript has significant merits, there are areas where improvements could further enhance its clarity, rigor, and impact.

I recommend the manuscript for publication with minor revisions, primarily focusing on clarifying specific points, adding relevant references, and refining certain sections to improve clarity and communication of the findings.

Minor comments:

1.Line 362-363 (Role of IL-11 in Decidualization):
The discussion on IL-11 requires clarification. The authors cite [75], which highlights that IL-11 is essential for mouse decidualization and that its receptor deficiency leads to infertility. However, they do not clearly articulate how these findings translate to the human context. It would be helpful to:

1) Contrast mouse and human data on IL-11 expression during decidualization.

2) Discuss potential mechanisms by which IL-11 dysregulation in eutopic endometrium might contribute to infertility in endometriosis patients.

2.Line 413-414 (PGE2 and EP Receptors):
The authors discuss the role of PGE2 and its receptors (EP2/EP4) during decidualization but fail to include primary references. Specific studies elucidating the role of PGE2 signaling in endometrial stromal cells during decidualization should be cited here. For instance:

1) Cite foundational work on the EP receptor pathways in endometriotic lesions and eutopic endometrium.

2) Discuss how these pathways might interact with other hormonal or inflammatory mediators in the diseased microenvironment.

3.Epigenetic Modifications (Line 390–447):
The section on epigenetic regulation of decidualization is well-written but could benefit from additional specificity. To improve clarity and accessibility: Provide a table summarizing key epigenetic marks (e.g., H3K27me3, H3K9me3) and their functional consequences on decidualization markers such as PRL and IGFBP1.

Suggestions for Improvement:

1.Introduction Section:
The introduction effectively sets the stage but could be refined to explicitly highlight the knowledge gap this review addresses. Specifically, the authors could emphasize the lack of understanding of the molecular basis of infertility in eutopic endometrium and how this review aims to bridge that gap.

2.Discussion Section:

The discussion provides valuable insights but could be expanded to explore the translational implications of the findings:

1) How could knowledge of progesterone resistance or epigenetic dysregulation inform the development of novel, targeted therapies?

2) Are there ongoing clinical trials or emerging therapies that align with the pathways discussed in this review? Addressing these questions would enhance the practical relevance of the manuscript.

3.Figures and Tables:
While the visuals are informative, some (e.g., Figure 2) are complex and could be simplified for broader accessibility. Additionally, providing more concise figure legends would make it easier for readers outside the immediate field to interpret the data.

Author Response

The manuscript titled "Molecular Basis of Impaired Decidualization in Eutopic Endometrium of Endometriosis Patients" is a well-structured and detailed review of an important and underexplored topic in reproductive biology. The authors have done an excellent job of integrating a broad range of current literature, presenting complex mechanisms in an accessible manner, and offering clinical and therapeutic perspectives. However, while the manuscript has significant merits, there are areas where improvements could further enhance its clarity, rigor, and impact.

 I recommend the manuscript for publication with minor revisions, primarily focusing on clarifying specific points, adding relevant references, and refining certain sections to improve clarity and communication of the findings.

Response: We thank the reviewer for recognizing the potential interest of our work. We appreciate the detailed and constructive critique that will help strengthen the manuscript. We have carefully revised the manuscript to address all comments and enhance its clarity, scientific rigor, and clinical relevance.

Minor comments:

1.Line 362-363 (Role of IL-11 in Decidualization):
The discussion on IL-11 requires clarification. The authors cite [75], which highlights that IL-11 is essential for mouse decidualization and that its receptor deficiency leads to infertility. However, they do not clearly articulate how these findings translate to the human context. It would be helpful to:

1) Contrast mouse and human data on IL-11 expression during decidualization.

2) Discuss potential mechanisms by which IL-11 dysregulation in eutopic endometrium might contribute to infertility in endometriosis patients.

Response: We thank the reviewer for this important point regarding IL-11 regulation in decidualization. We apologize for any confusion in our previous discussion of the Robb et al. study. We have thoroughly revised this section to provide a clearer comparison between mouse and human IL-11 expression during decidualization (LINES 357-369). Specifically, we have:

  1. Clarified the species-specific differences in IL-11 regulation during decidualization
  2. Expanded the discussion of IL-11's role in both mouse and human decidualization
  3. Added a detailed analysis of how reduced IL-11 expression in eutopic endometrium (likely resulting from progesterone resistance) may contribute to infertility through multiple mechanisms including:
  • Impaired decidualization
  • Compromised implantation
  • Altered placentation
  • Disrupted Natural Killer cell differentiation
  • Reduced survival of decidualized cells

These modifications provide a more comprehensive understanding of IL-11's role in endometriosis-associated infertility in the eutopic endometrium.

2.Line 413-414 (PGE2 and EP Receptors):
The authors discuss the role of PGE2 and its receptors (EP2/EP4) during decidualization but fail to include primary references. Specific studies elucidating the role of PGE2 signaling in endometrial stromal cells during decidualization should be cited here. For instance:

1) Cite foundational work on the EP receptor pathways in endometriotic lesions and eutopic endometrium.

2) Discuss how these pathways might interact with other hormonal or inflammatory mediators in the diseased microenvironment.

Response: We thank the reviewer for this valuable suggestion regarding PGE2 signaling. We have revised the manuscript to include: Several key foundational and recent data describing PGE2 pathway regulation during decidualization, evidence for PGE2 signaling dysregulation in endometriosis, and a discussion of the interactions between PGE2 signaling and other pathological mediators in the disease (LINES 437-457).

3.Epigenetic Modifications (Line 390–447):
The section on epigenetic regulation of decidualization is well-written but could benefit from additional specificity. To improve clarity and accessibility: Provide a table summarizing key epigenetic marks (e.g., H3K27me3, H3K9me3) and their functional consequences on decidualization markers such as PRL and IGFBP1.

Response: We thank the reviewer for this constructive suggestion to improve the clarity of our epigenetic regulation section. We have:

  1. Expanded Table 3 and the manuscript (LINES 418-426) to include specific histone modifications and their regulatory effects on decidualization markers.
  2. Added recent data describing hypoxia-induced epigenetic regulation in endometriosis. Specifically, we included information about EZH2 mRNA stabilization through m6A modification and the consequent maintenance of H3K27me3 at the IGFBP1 promoter and its impact on decidualization capacity (LINES 427-436).

These modifications provide a more comprehensive and accessible overview of epigenetic regulation in endometriosis-associated decidualization defects.

Suggestions for Improvement:

1.Introduction Section:
The introduction effectively sets the stage but could be refined to explicitly highlight the knowledge gap this review addresses. Specifically, the authors could emphasize the lack of understanding of the molecular basis of infertility in eutopic endometrium and how this review aims to bridge that gap.

Response: We thank the reviewer for this valuable suggestion to strengthen our introduction. We have revised the introduction to better highlight the current knowledge gap regarding molecular mechanisms underlying endometriosis-associated infertility.

Specifically, we have added the following paragraph (LINES 69-76):

"While endometriosis-associated infertility is well documented, the molecular basis of eutopic endometrial dysfunction remains incompletely understood. This review synthesizes current evidence from both in vivo and in vitro studies examining decidualization defects in eutopic endometrium of patients with endometriosis. By integrating recent molecular and cellular findings, we aim to provide comprehensive insights into this underexplored aspect of reproductive biology and identify potential therapeutic targets for improving fertility outcomes in endometriosis patients."

2.Discussion Section:

The discussion provides valuable insights but could be expanded to explore the translational implications of the findings:

  • How could knowledge of progesterone resistance or epigenetic dysregulation inform the development of novel, targeted therapies?

Response: We appreciate the interesting question raised by the reviewer. Accordingly, we have included the following information in the new version of the manuscript:

LINES 203-213: “Enhanced understanding of endometriosis-associated molecular mechanisms provides crucial insights for developing hormone signaling-targeted therapies. Specifically, elucidating progesterone resistance mechanisms will facilitate the development of selective progesterone receptor modulators that account for PGR isoform-specific alterations and epigenetic regulation of receptor expression. Novel therapeutic strategies targeting membrane progesterone receptors could include PAQR family-specific modulators and pathway-selective interventions. Additionally, interventions aimed at normalizing estrogen signaling through ESR1/ESR2 ratio modulation, regulation of local estrogen production, and integration with progesterone signaling pathways represent promising therapeutic approaches. These mechanistic insights will enable the development of more targeted and effective treatments for endometriosis-associated infertility.”

LINES 537-542: “Understanding these molecular mechanisms will suggest several promising therapeutic strategies, including targeted epigenetic interventions to modulate DNA methylation, regulation of histone-modifying enzymes, and development of selective EP2/EP4 receptor antagonists to normalize histone modifications. Future therapeutic approaches could incorporate tissue-specific delivery systems to maximize efficacy while minimizing off-target effects.”

2) Are there ongoing clinical trials or emerging therapies that align with the pathways discussed in this review? Addressing these questions would enhance the practical relevance of the manuscript.

 Response: We thank the reviewer for this valuable suggestion to enhance the clinical relevance of our review. We have added a new section (7) titled "Therapeutic Approaches Targeting Dysregulated Pathways in Eutopic Endometrium of Endometriosis Patients" (LINES 544-575) that briefly discusses current therapeutic strategies and emerging epigenetic therapies. This section discusses the therapeutic potential of targeting DNA methylation patterns, histone modifications, and hormone signaling pathways, while analyzing challenges and opportunities in developing targeted therapies for endometriosis-associated infertility.

3.Figures and Tables:
While the visuals are informative, some (e.g., Figure 2) are complex and could be simplified for broader accessibility. Additionally, providing more concise figure legends would make it easier for readers outside the immediate field to interpret the data.

Response: We thank the reviewer for these constructive suggestions to improve the figures' clarity and accessibility. We have implemented the following modifications:

We have simplified Figure 1 to enhance its clarity. In panel A, we streamlined the presentation by using 'DAG' to represent decidualization-associated genes and replaced the cross symbol with 'impaired decidualization' to better illustrate PGR expression effects. Panel B has been extensively revised to focus solely on PAQR expression changes, removing redundant protein-level details. The original panel C showing steroidogenesis pathways has been removed, and the former panel D is now presented as the new panel C, resulting in a more focused and accessible figure.

Figure 2 has been redesigned to improve visual clarity while maintaining essential mechanistic information. We have streamlined the layout by simplifying the visual elements and reducing directional arrows. A unified arrow system now indicates expression and protein level changes during decidualization, while the representation of SMAD and BMP signaling pathways has been condensed to show only key regulatory steps. Both pathways are now integrated at the FOXO1 locus despite their independent regulation. In panel B, SMAD and BMP signaling pathways have been omitted to accurately reflect their disruption during decidualization in ectopic endometrium.

Additionally, we have revised all figure legends to be more concise and accessible.

Reviewer 2 Report

Comments and Suggestions for Authors

This review article summarizes recent reports on the mechanisms by which impaired decidualization is involved in infertility due to endometriosis. Overall, the article is well-organized and well-written. However, some parts are quite difficult for readers to understand, and revisions are necessary.

Specific Comments:

1.     Figure 1: The figure is large but unclear. It lacks focus, and the key points are not easily conveyed due to issues with font size and layout. Avoid including text within the figure.

2.     Focus on Eutopic Endometrium in the patient: As the title suggests, the review should focus on the eutopic (properly located) endometrium. However, the content involves complex discussions along with changes in ectopic endometrial tissue. In this regard, Table 1 should only describe the comparison between eutopic and normal endometrium, and the 'Results' should be clearly stated. The same applies to Table 2.

Minor Comments:

1.     In section #3 on ‘hormonal signaling~’, the discussion actually pertains to ovarian steroids and sex hormones. These hormones are also related to IGFBP1/PRL, which is addressed in section #4, and function downstream of them. It might be more accurate to replace "Hormonal" with "ovarian steroids" or similar terms.

2.     Definition of Eutopic Endometrium: The eutopic endometrium refers to the normal endometrial tissue located in the proper place in the uterus. In contrast, the ectopic endometrium refers to endometrial tissue located outside the uterus. Thus, while the eutopic endometrium refers to the patient's normal endometrial tissue, can the endometrium of a healthy person be called eutopic?

3.     Lines 115-116: The phrase "PGRMC, progestin receptor?" is unclear and should be clarified.

4.     Lines 241-244: This section should be described more clearly.

5.     Lines 449-451: A reference should be inserted.

Comments on the Quality of English Language

I am not native english speaker.

Author Response

This review article summarizes recent reports on the mechanisms by which impaired decidualization is involved in infertility due to endometriosis. Overall, the article is well-organized and well-written. However, some parts are quite difficult for readers to understand, and revisions are necessary.

Response: We appreciate the detailed and constructive critique that will help strengthen the manuscript. We have carefully revised the manuscript to address all comments and enhance its clarity and organization.

Specific Comments:

  1. Figure 1: The figure is large but unclear. It lacks focus, and the key points are not easily conveyed due to issues with font size and layout. Avoid including text within the figure.

Response: We thank the reviewer for these constructive suggestions to improve the figure 1 clarity. Accordingly, we have simplified Figure 1 to enhance its clarity. In panel A, we streamlined the presentation by using 'DAG' to represent decidualization-associated genes and replaced the cross symbol with 'impaired decidualization' to better illustrate PGR expression effects. Panel B has been extensively revised to focus solely on PAQR expression changes, removing redundant protein-level details. The original panel C showing steroidogenesis pathways has been removed, and the former panel D is now presented as the new panel C, resulting in a more focused and clearer figure.

  1. Focus on Eutopic Endometrium in the patient: As the title suggests, the review should focus on the eutopic (properly located) endometrium. However, the content involves complex discussions along with changes in ectopic endometrial tissue. In this regard, Table 1 should only describe the comparison between eutopic and normal endometrium, and the 'Results' should be clearly stated. The same applies to Table 2.

Response: We thank the reviewer for this important observation regarding the manuscript's focus. While our primary emphasis is on eutopic endometrium, we believe that including findings from ectopic lesions provides valuable context for understanding molecular alterations in eutopic tissue. This comparative approach is particularly relevant given that in some cases molecular mechanisms are shared between eutopic and ectopic endometrium, the current literature predominantly focuses on ectopic lesions, and recognizing ectopic tissue alterations can help to understand the eutopic endometrium pathology.

Nevertheless, we have revised the manuscript thoroughly to better emphasize our focus on eutopic endometrium by:

  • Modifying Tables 1 and 2 to specifically compare eutopic endometrium from endometriosis patients with normal endometrium.
  • Clarifying when findings from eutopic tissue are being discussed.
  • Highlighting the need for more studies focused on eutopic endometrium.

Minor Comments:

  1. In section #3 on ‘hormonal signaling~’, the discussion actually pertains to ovarian steroids and sex hormones. These hormones are also related to IGFBP1/PRL, which is addressed in section #4, and function downstream of them. It might be more accurate to replace "Hormonal" with "ovarian steroids" or similar terms.

Response: According to the reviewer suggestion, we have modified the title of section #3 to “Sex-Steroid Hormonal Signaling, Dysregulation and Impaired Decidualization in Endometriosis”.

  1. Definition of Eutopic Endometrium: The eutopic endometrium refers to the normal endometrial tissue located in the proper place in the uterus. In contrast, the ectopic endometrium refers to endometrial tissue located outside the uterus. Thus, while the eutopic endometrium refers to the patient's normal endometrial tissue, can the endometrium of a healthy person be called eutopic?

Response: We thank the reviewer for raising this important point regarding terminology. We acknowledge that while "eutopic" technically refers to "occurring in the normal position," usage in endometriosis research has evolved. In the field, "eutopic endometrium" commonly refers to properly located uterine endometrium specifically in endometriosis patients, while endometrium from women without the disease is designated as control or normal endometrium. We have clarified this terminology (lines 65-66 and 105-109) and revised the manuscript throughout to maintain consistent distinction between eutopic endometrium from endometriosis patients and control endometrium from healthy individuals.

  1. Lines 115-116: The phrase "PGRMC, progestin receptor?" is unclear and should be clarified.

Response: We apologize for the incorrect nomenclature regarding progesterone membrane receptors. We have revised their terminology throughout the manuscript to ensure accuracy and consistency with current nomenclature (lines 125-127).

  1. Lines 241-244: This section should be described more clearly.

Response: We thank the reviewer for highlighting this clarity issue. We have revised the section (lines 261-265) to read:

"In ESCs isolated from both endometriosis patients and controls, P4 treatment (14 days) upregulates IGFBP1 and PRL expression. However, when treated with 8-Br-cAMP (96 hours), control ESCs show significantly higher expression of both markers compared to endometriotic ESCs [49], suggesting PKA pathway involvement in this differential response."

  1. Lines 449-451: A reference should be inserted.

Response: According to the reviewer’s comment, we have included the corresponding reference (LINE 489).

Round 2

Reviewer 2 Report

Comments and Suggestions for Authors

The authors have appropriately corrected the manuscript according to our comments. 

Author Response

Comments: The authors have appropriately corrected the manuscript according to our comments.  Response: We sincerely thank the reviewer for the constructive feedback and thorough evaluation that helped improve our manuscript.